
# The Impact and Resolution of the GPS Week Number Rollover of April 2019 on Autonomous Geophysical Instrument Platforms

Shane Coyle[1], C. Robert Clauer[1,2], Michael D. Hartinger[3], Zhonghua Xu[1,2], and Yuxiang Peng[1]

[1]Virginia Polytechnic Institute and State University, Blacksburg VA, US
[2]National Institute of Aerospace, Hampton VA, US
[3]Space Science Institute, Boulder CO, US

**Correspondence:** Shane Coyle, Center for Space Science and Engineering Research, Bradley Department of Electrical and Computer Engineering, Virginia Tech, Blacksburg, VA 24061, USA. (shanec1@vt.edu)

**Abstract.** Instrument platforms the world over often rely on GPS or similar satellite constellations for accurate timekeeping and synchronization. This reliance can create problems when the timekeeping counter aboard a satellite overflows and begin a new epoch. Due to the rarity of these events (19.6 years for GPS), software designers may be unaware of such circumstance, or may choose to ignore it for development complexity considerations. Although it is impossible to predict every fault that

may occur in a complicated system, there are a few "best practices" that can allow for graceful fault recovery and restorative action. These guiding principles are especially pertinent for instrument platforms operating in space or in remote locations like Antarctica, where restorative maintenance is both difficult and expensive. In this work, we describe how these principles apply to a communications failure on Autonomous Adaptive Low-Power Instrument Platforms (AAL-PIP) deployed in Antarctica. In particular, we describe how code execution patterns were subtly altered after the GPS week number rollover of April 2019,

how this led to Iridium satellite communications and data collection failures, and how communications and data collection were ultimately restored. Finally, we offer some core tenets of instrument platform design as guidance for future development.

## 1 Introduction

### 1.1 GPS

In the 1970's, the United States Department of Defense (DoD) initialized and led the Navigation System with Timing and
Ranging (NAVSTAR) program in order to establish a robust, stable, global satellite navigation system. Since the launch of the Navstar-1 satellite in 1978, a constellation of 24 early NAVSTAR satellites became the original space segment of fully operational Global Positioning System (GPS) around 1993. Owned by the US Government and operated by the United States Air Force, GPS provided two levels of service: Standard Positioning Service (SPS) and Precise Positioning Service (PPS). Also known as the military GPS service, the PPS was restricted to authorized users only (e.g. US Armed Forces, US Fed-
eral agencies). The SPS performance was degraded intentionally via a program called "Selective Availability (SA)". SA was disabled in May 2000 by President Bill Clinton, which marked the era of scientific, civilian, and commercial usage of GPS. The positioning, navigation and timing (PNT) applications using GPS include (but are not limited to) astronomy, autonomous





systems (Peng et al., 2020), geo-science, space weather (Peng and Scales, 2019; Peng et al., 2019), survey, disaster services, object tracking, etc.

As of November 2020, there are 31 GPS space vehicles (SV) operational in 6 different orbital planes with 60 degree separation angles in between. All SVs are in medium Earth orbits (MEO) with an altitude of approximately 20200 km. Each GPS SV orbits the Earth about twice a day (orbital period = 11 hours 58 minutes) in a near circular orbit (eccentricity < 0.02). In an open sky environment, in principle, at least 8 GPS SVs can be simultaneously seen at anywhere on the Earth's surface at given time (Rizos et al., 2010). Although the positioning accuracy depends on GPS receiver design and signal environment, it is

expected to achieve horizontal accuracy of 3 meters or better and vertical accuracy of 5 meters or better 95% of the time for well-designed GPS receivers (Moorefield, 2020). Thanks to the atomic clocks on GPS satellites, the GPS timing accuracy is typically in the 1 to 10 ns range (Allan and Weiss, 1980). Because the clock on GPS receivers (usually a crystal oscillator) is much less robust then the GPS satellite clock, a receiver clock offset needs to be estimated in solving a GPS receiver's position using Kalman filters or other estimation techniques. The GPS timing system consists of two parameters: GPS week and GPS

seconds of the week. The weekly counter is stored with a total of ten binary digits, therefore the GPS week number rollovers every 1024 weeks. Since the initial GPS epoch is 0h UTC (midnight) of January 6 to 7 in the year of 1980, the first GPS rollover occurred on midnight (UTC) of August 21 to 22 in the year of 1999. The second rollover took place on midnight of April 6 to 7 in the year of 2019. GPS is not the only Global Navigation Satellite System (GNSS) that experiences a periodic rollover event. The European Union's Galileo network uses a 12 digit week number, and will rollover in February 2078. Similarly, the

next rollover event for the 13 digit week number of the Chinese Beidou network is in January 2163 (ublox, 2020).

Although the 2019 GPS rollover was warned of by Department of Homeland Security, the International Civil Aviation Organization and other relevant organizations beforehand, numerous events are known to be impacted by the 2019 GPS rollover. For example, numerous airlines flights were delayed or cancelled due to a failure of the airplane's flight management and navigation software caused by the 2019 GPS rollover (Gallagher, 2019). Other events include the crash of the New York City

Wireless Network (Divis, 2019), the communications systems failure of Australian Bureau of Meteorology's weather balloons (Cozzens, 2019), etc. It is not uncommon for the impact of clock faults to be subtle or take effect after some delay, as is the case in the fault described here.

## 1.2 AAL-PIP

The Autonomous Adaptive Low-Power Instrument Platforms (AAL-PIP) are one example of a civilian application of GPS, as

they rely on GPS for accurate timekeeping and synchronization. The AAL-PIPs are used for space weather research, and they consist of several instruments that make a variety of measurements: fluxgate magnetometer, search coil magnetometer, dual-frequency GPS receiver, and a high-frequency radio experiment. Six AAL-PIPs are currently deployed in Antarctica between approximately -82 and -85 degrees latitude, with two more planned deployments east of the present meridional chain. Data from these AAL-PIPs, combined with data from a magnetically conjugate chain of stations in the northern hemisphere, are

currently being used to study north-south hemisphere differences in the coupling between the solar wind, magnetosphere, and ionosphere (Clauer et al., 2014; Xu et al., 2019).


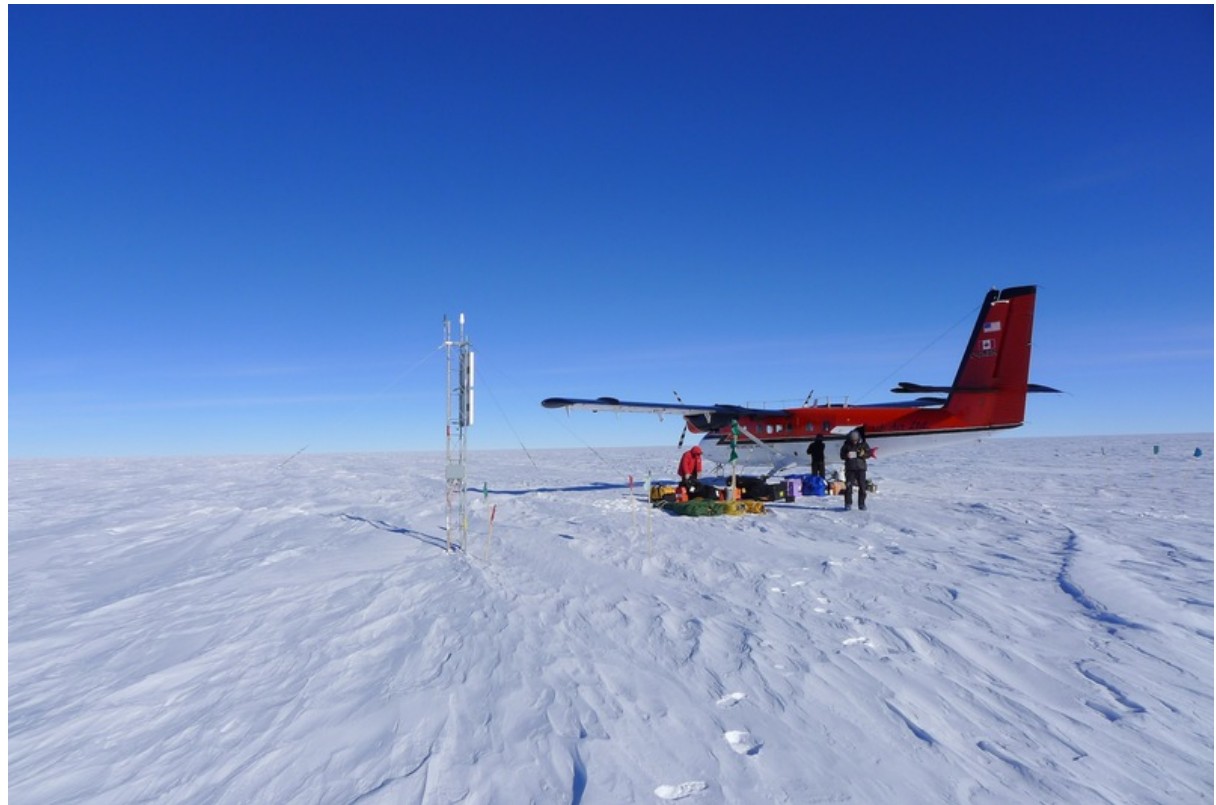

**Figure 1.** Twin Otter airplane next to the AAL-PIP system at PG2 (84.42 S, 57.96 E). The extreme environment and remote nature of the AAL-PIP sites makes in-person maintenance very difficult.

Deployment and maintenance visits to the Antarctic AAL-PIPs are challenging. For example, in order to install the instrument platforms, two trips were required for each field site, one for the equipment cargo and another for the field team to put in. For several of the sites, the distance required the establishment of fuel caches at intermediate locations and at the site, which required additional flights. Weather and visibility present additional complications. With the high demand on aircraft and the limited time that flying conditions are sufficient, it is sometimes the case that only one station can be visited during a particular summer field season. It is also noteworthy, that if a problem is detected at a station that requires a maintenance trip, bringing the equipment back for diagnosis and repair and then redeployment generally requires a year since it is only possible to visit during the summer field season.

Since access to remote locations in Antarctica is challenging, the AAL-PIPs are designed to operate unattended for five or more years. In general, they have met this expectation, with some systems operating for more than 12 years with no site visits or maintenance (e.g., PG1 system deployed in 2008, Hartinger et al. (2017)). Specific design features to reduce the need for onsite maintenance include a reliance on solar panels and batteries with no moving parts and Iridium 2-way communication to enable remote operations and maintenance. In the polar region, the available sunlight is limited throughout the year, and

so the systems eventually run out of power and gracefully enter hibernation during the austral winter. The time frame for this
hibernation period varies by site location and weather, but typically begins around August and ends in October. It is customary
to reduce the amount of data being retrieved on a daily basis to only important housekeeping values once the sun retreats in the
northern spring to extend the operational period. Science data is stored in local memory for later download when solar power
is plentiful.

The AAL-PIPs that are the focus of this paper used the Router-based Unrestricted Digital Interworking Connectivity Solutions
(RUDICS) Iridium protocol for communications. With RUDICS, a remote AAL-PIP system connects to the Iridium network
with its modem, and communications traffic is routed through a Department of Defence gateway in Hawaii. The gateway in
turn connects via the internet (Transmission Control Protocol/Interenet Protocol, or TCP/IP) to a server at Virginia Tech.
The GPS week number rollover of 2019 created an unanticipated software fault that affected both data collection and Iridium
communications at the AAL-PIPs. In this paper, we describe why this fault occurred, how it was diagnosed, and how it was
ultimately resolved. Thus, we use this fault as an example to describe best practices for designing future systems against similar
faults, or to at least enable graceful fault recovery and restorative action if a fault occurs.

Section 1 (this section) establishes a sufficient background in the operation and usage of satellite timing systems (Section
1.1), as well as details about the remote instrument platform operated by the authors (Section 1.2). Section 2 describes the onset
of the fault as discovered by the operators. Section 3 describes the development and implementation of a recovery solution,
and Section 4 lists some of the valuable insights gained from this effort. A summary is provided in Section 5.

## 2   Fault Description

In the case described below, we describe a fault that appears very minor at first which months later becomes a major fault that
halts operation at 4 of 6 stations. The fault encountered on the AAL-PIPs relates to the nature of clocks on Linux systems.
The single board computer at the heart of the AAL-PIP platform runs a very lightweight version of Linux based on the Debian
distribution. Like all Linux systems, there are two clocks running during operation. The "hardware" clock is typically used
when a device is "powered off" to keep track of time elapsed between reboot cycles and often relies on a coin cell battery to
maintain operation. The "system" clock is the clock that users are more familiar with, and represents the time and date with
a software counter. These clocks drift at different rates, and there are a number of different techniques to synchronize the two
over long periods of time.

For the purposes of operating the AAL-PIP platform, the system clock is only synchronized one time after a power outage or
reboot. To compensate for drift, GPS time is used to make subsequent adjustments to the system clock thereafter. The drift rate
measurements and compensations occur hourly. The distinction between clocks and the details of the adjustment process are
important for illustrating the complex behavior relating to the GPS fault on the AAL-PIP systems.


## 2.1 GPS Rollover, April 2019

On 6 April, 2019, it was noted that the system at Antarctic site PG2 had not completed its regular daily download. After consulting the server logs, it was discovered that the system in question had failed to connect to the server and had timed out of its download window. The following day, none of the systems were able to connect, and it was discovered that there were several dozen stalled ssh sessions in the server process listing. After clearing the hung processes and restarting the proxy software normal operations were resumed. The server software has a known bug which can result in an incomplete disconnect from an ssh session, so this behavior was believed to be simply the result of an accumulation of these failed attempts. The coincidence with the GPS rollover was noted and suggested as the probable cause. However, because operations were able to resume as normal after restarting the server software, little further consideration was given.

## 2.2 Season Commencement, November 2019

As previously explained, the typical time frame for AAL-PIP systems to begin operating after hibernation is in late October to early November. By mid-November, none of the RUDICS style systems had successfully begun regular communication with the base station server. Inspecting the logs found a few instances at the beginning of the season where the systems were able to make first contact, but this activity lapsed after just several days. Up to this point 0 of the 5 RUDICS enabled stations were able to send home housekeeping data. Eventually one system established regular connections, but it was clear at this point that a serious failure had occurred.

One of the capabilities afforded by using a satellite communications network like Iridium is the ability to determine whether or not a device has established a link to a satellite, regardless of the state of the end-to-end communication link. By contacting the network operator and accessing the call logs, it was determined that the systems were indeed powered up and attempting to call the server. Further analysis of the TCP traffic incident from the RUDICS gateway coordinated with call logs matched to each system indicated that the fault had occurred in the handshaking routine between the server and the remote system.

Along with reaching out to the network controller for assistance, efforts were made to replicate the fault in a bench-top development system that parallels the hardware deployed in the field. While it is not always feasible to have a reserved set of instruments available for troubleshooting, many devices can be simulated with readily available electronics test equipment. This is particularly true of the AAL-PIP platforms, on which instruments are digitized by hardware located in the electronics box and not on the sensors themselves.

During simulated operational testing of the hardware onsite in Blacksburg, VA, it was determined that the GPS module was providing an erroneous date and time. A constant negative offset of 2048 weeks was found, which is consistent with the GPS rollover period. Further analysis discovered the source of the communications fault, which occurs as a result of updating the system time while an active communications link is established. A communications watchdog is supposed to monitor unusual circumstances relating to "hung" or stalled communications links, but the watchdog relies on the system clock and is effectively disabled because of the bug. The process of determining how much time has elapsed between successful communications attempts is a simple subtractions of a watchdog start time from the current system clock time. However, when the system time





went through its update process, it set the system clock back 19 years, and the subtraction to determine the elapsed time yielded
a negative result. In the case of the AAL-PIP watchdog, an elapsed time of two hours without successful communications will
trigger the watchdog. For the bugged negative result to ever reach that threshold after the fault occurs would take 19.6 years
and two hours.

## 3   Fault Recovery

Determining the source of the fault was a crucial first step in restoring the operational status of the AAL-PIP array. Following
the fault isolation, it was necessary to develop a software patch that addressed the rollover related time setting bug. There were
two important issues to resolve in writing any potential solutions. First, because of differences in GPS firmware, determination
of whether the bug was present had to take place. It was noted that one remote system had not entered the fault state, and it is
suspected that this is because it was built several years after the initial platforms. Likely, this later GPS module had a firmware
update that could handle the rollover. Isolating fault occurrence was achieved by first determining if the reported date was prior
to the current GPS epoch start date. Second, if the date was incorrect, the reported difference in date and time had to be offset,
in this case by simply adding a constant value to the recovered value. This software patch was developed in December of 2019
in advance of the austral summer field season. Figure 2 shows the differences in timing information flow within the AAL-PIP
systems as a result of this patch. The first attempt to deploy this fix would require a manual power cycle of the remote system,
something not achievable without ground support.

### 3.1   Repair Activity, January 2020

Travelling to remote locations on the Antarctic plateau requires significant expenditures of resources as well as cooperation
from the weather. After several canceled attempts to fly out to the field, a team was finally able to land at PG0 on January
6th, 2020. Weather and fuel conditions only allowed for a short visit of around 30 minutes. As this allotment of time would
not facilitate digging out the electronics box to disconnect the GPS cable, the field team instead attempted to climb the solar
panel tower and disconnect the GPS antenna. Years of wear and frigid temperatures presented challenges in disconnecting the
antenna, and eventually the decision was made to abandon this effort. A final attempt to attenuate the GPS signal by shielding it
proved equally in vain. As a last ditch effort to restore communications to the system, the field team was advised to cut the GPS
antenna cable at the base of the tower. After manually cycling power, the team left the site. The system at PG0 successfully
established communications with the server before the field team landed back at South Pole Station, thus proving beyond doubt
that the GPS module was the source of the communications fault.
With communications restored, updates to the system software could now be deployed in order to restore normal operation.
After several bouts of local testing, a final updated version of the software that addressed the GPS fault was installed on January
15. Because this system no longer had a functional GPS antenna, the software was modified to record data regardless of the
status of GPS. In the event of the GPS module failing to provide a sensible date/time, the IRIDIUM network of satellites would
provide a usable reference clock. The hardware clock drift measurements would then be made using the IRIDIUM network





clock, and any offsets could be recorded and adjusted as a post-processing step. Data from PG0 have been artificially corrected in this way since January 2020, and the data compare well with measurements from other local stations.

## 3.2 Repair Activity, October 2020

The efforts of the field team had demonstrated that future updates to the software running on the remote systems would restore
the operational capacity of the platforms. Unfortunately, the process of updating that software required manually power cycling the systems. After the first site visit, no future visits were deemed possible for the 2019/2020 season. This meant any future repair efforts were delayed until late 2020 at the earliest. Amidst the global pandemic and other circumstances surrounding Antarctic field support that year, it seemed unlikely that sufficient resources would be allocated to support site visits until the following season (2021/2022) at the earliest.

During the northern summer of 2020, it was suggested that there may be a small window of opportunity to deploy the software updates remotely at the beginning of the season. Because the AAL-PIP systems go into a hibernation period (typically in July-August) there is a natural power cycling that occurs during that period. If a script could be written to "catch" that first period of attempted connections to the server at Virginia Tech, it might be possible to interrupt the GPS driver. This would prevent the platform from recording data, but it would also prevent it from entering the fault state. This would allow a further
software update to fix the GPS bug and prevent any future failures. Work on this script completed in August, and tests showed that the total time to detect a TCP/IP connection, establish a socket, create an SSH session, and then run the script was about 70 seconds. The window for getting a GPS sync can vary significantly based on location and power availability, but it was estimated to be about 90 seconds in this case, affording us a 20 second buffer. The script was written to poll for an established connection every 2 seconds, and was set to run in early October, 2020.

On October 12, the first of the faulted systems attempted to call home and establish regular communications. The script ran as intended, immediately logging in to the remote system and both disabling and uninstalling the GPS driver. A subsequent software update was pushed to the remote system, and normal operations were restored at that site. At the time of this writing, all but one of the systems have come out of hibernation and each system has undergone a software update. The AAL-PIP array has thus far been restored to full operational capacity.

## 190 4 Lessons Learned

There are a number of broadly applicable lessons to be learned from the AAL-PIP GPS fault of 2019 and subsequent fix. Arguably the most important contribution to our success was the ability to deliver over-the-air updates through the IRIDIUM satellite network. Another takeaway is the importance of watchdog design, and more broadly software design in general. Finally, a more thorough consideration of the implicit and explicit assumptions that arise during instrument development can
prevent these kinds of faults from occurring in the future.





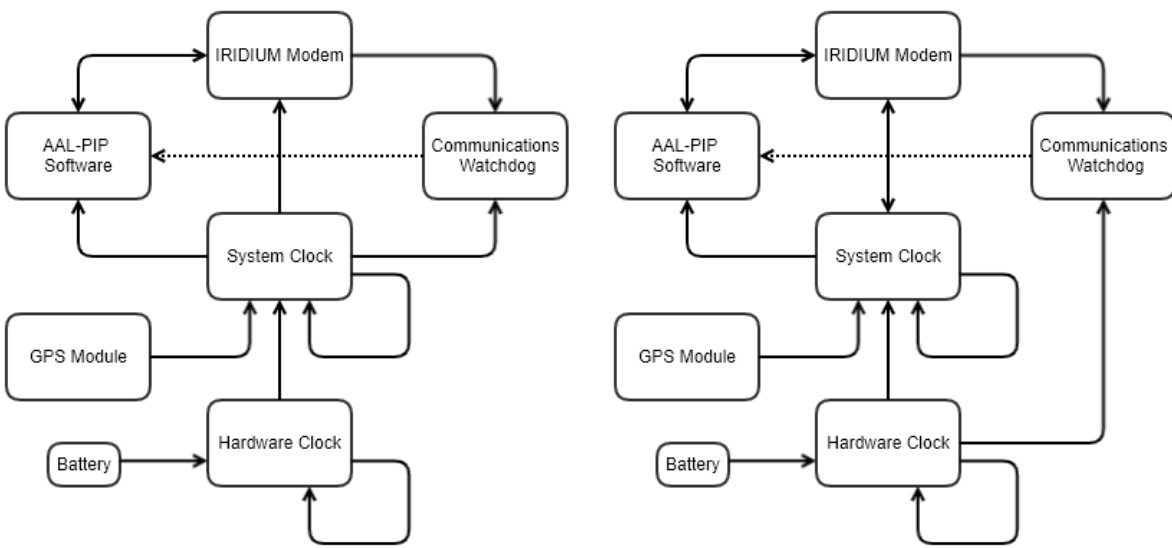

**Figure 2.** Diagrams contrasting the original flow of time information on the AAL-PIP systems (left) with the current method (right). Note the use of the hardware clock as an input for the communications watchdog, as well as the availability of IRIDIUM network sourced time for the system clock.

## 4.1 Remote Communications

Without the ability to interact with the remote platforms, these sites would likely have remained non-functional for another year or two. The ability to remotely diagnose and operate systems is paramount, and should be of the utmost consideration during development of instrument platforms. Any software updates that are written must be verified first and foremost to not

disable the system's ability to establish a communications link with the operator. When operating an instrument platform in remote locations, the cost of maintenance is directly tied to how often the site must be visited, and the most effective way to reduce this cost is to eliminate the need to visit the site. For AAL-PIP, this starts by providing our own power through solar panels, and is sustained by using the IRIDIUM satellite network and RUDICS communications (Clauer et al., 2014).

A robust software and hardware design using a single board computer (SBC) and a common Linux operating system (Debian)

make this kind of fault recovery possible on the AAL-PIP systems. As microcomputers become ever smaller and more cost effective, they become a more appealing solution to development of instrumentation platforms. Indeed, future iterations of the AAL-PIP platform that are soon to be deployed in cooperation with the Polar Research Institute of China (PRIC) utilize an updated version of the SBC found in the first generation. Some might consider going to an even smaller and less power hungry platform for instrument control like that afforded by a microcontroller without all of the peripherals found in a microcomputer.

Whichever control platform is chosen, it is important to consider and plan for the eventuality that updates and fixes may need to be deployed to the system.





## 4.2 Clocks and Watchdogs

In the case of the AAL-PIP system, there are two important software design considerations that had to be addressed during the patch process. First, the communications watchdog timer issue had to be resolved. Typically, the watchdog will automatically
reboot the system after either staying connected or disconnected for more than 24 hours. When the watchdog first starts, it obtains the current time and then sets a trigger for 24 hours later. This reboot process was prevented by rolling the clock back to a time several years before the trigger time. A different, more robust solution to using the full date & time for timers like this would have been to strip the date from the trigger. However, this can still lead to some confusion as a result of making adjustments to the system clock. Using a monotonic hardware clock for this timer would provide a more robust solution, as
these clocks typically are not adjusted by any software process.

## 4.3 Checking Assumptions

The solution described in the Fault Recovery section involved checking the reported time against some epoch, and only adding an offset if the reported time was incorrect based on that comparison. This solution will only work until the next GPS rollover in 2038. Ideally, the systems will go through a hardware refurbishment before then, and newer GPS modules using the 13
digit standard can be installed. This rollover won't occur until 2137, and is sufficiently beyond consideration here. However, a rollover of a different kind will also occur in 2038, that of the signed integer time using the Unix epoch date of 1970. Though an entire adolescence will occur between now and that point, it is never too soon to consider what problems may arise as a result.

It is assumed that clocks are a relatively trivial device inside of a computing system, and that the reported time can be taken for
granted. Certainly for most applications, especially those using a network for timing, this is typically true. However we have shown here that it is not always the case and can have dramatic impacts on system performance. It was originally assumed that the GPS time reported would always be accurate to within milliseconds, and this assumption was written into our software. When the time was offset by 1024 weeks, our system performance suffered as a result. In this instance, it was worth considering that the reported time could have been incorrect, and planning for that failure to relay the correct time. Our software was able
to account for the situation where no GPS time was reported, but not for incorrect GPS time reported.

## 5 Summary and Conclusions

To summarize our suggested design guidelines:

– Maintaining control of the system behavior through use of a bidirectional communications link is of the utmost importance for system maintenance and repair.

– Reliance on software based timing is not recommended for subsystems critical to instrument platform operation. Monotonic hardware timers should be used for watchdogs and other critical timers whenever possible.

- It is important to consider both hard and soft subsystem failures, where outputs may be either missing or invalid.

- Special consideration during the design phase should be given to identifying and reducing the number of single point failure mechanisms. This includes any subsystem that could fail in a way that eliminates control of the system as a whole.

- Integration testing should be extensive, and attention should be given to potential faults in each subsystem.

- Whenever possible, duplicate hardware setups should remain onsite at the operational facility to develop solutions to unforeseen faults. If this is considered infeasible, some other method of simulating system behavior (through software / virtual instruments) should be considered.

Operating an instrument platform in a remote location like Antarctica provides a unique set of challenges for maintenance and
operations. When setting out to develop an new remote instrument project or refurbish an existing project, there are a number of guiding principles one can follow to ensure success. Maintaining a robust communications link with the platform ensures that any eventualities that may arise from unsuspected fault cases can be addressed remotely. This ensures the platform can be maintained at minimal cost and expenditure, prolonging the life of the instrument and leading to greater proliferation. Properly identifying potential faults and adding fail-safes to critical systems is paramount to achieving this low-cost, long-life status.
Careful consideration must be given to how a system reacts in the event of hard failures as well as soft failures; the difference between a module not working instead of working incorrectly can cause dramatically different behaviors. This is exactly the sort of situation that occurred on the AAL-PIP platforms as a result of the GPS rollover in 2019, and this kind of situation may occur twice in the year 2038. With thoughtful attention to how a platform will operate in the event of a failure, instrument designers and operators can plan for and recover from even the most unexpected and thorough fault cases.

*Author contributions.* SC currently maintains the AAL-PIP array operations and has prepared the manuscript with assistance from all co-authors. SC is also responsible for the development of software updates previously mentioned. YP provided troubleshooting assistance by operating a GPS synthesizer during fault replication attempts. CRC, MH, and ZX are all members originally involved in the deployment of the AAL-PIP magnetometer array.

*Competing interests.* The authors declare they have no conflict of interest.

*Acknowledgements.* This material is based upon work supported by the National Science Foundation under Grant No. PLR-1543364, AGS-2027210, and AGS-2027168. The authors acknowledge and appreciate the assistance provided by USAP, specifically attributing the efforts of Dan Wagster to the successful diagnosis of the communications fault. John Bowman, and undergraduate student at Virginia Tech is also credited with assistance during the fault diagnosis. The New Jersey Institute of Technology (NJIT) is credited with providing Antarctic field



personnel and expertise during field work at PG0.




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
