# Peer review of "The Impact and Resolution of the GPS Week Number Rollover of April 2019 on Autonomous Geophysical Instrument Platforms"

_Geoscientific Instrumentation, Methods and Data Systems, 2020_

## Author Response (AR1)

==== Minor edits ===

>>line 35 "therefore the GPS week number rollovers every 1024 weeks"

...number rolls over every...

**Commented [SC1]:** done

>>line 41 "Although the 2019 GPS rollover was warned of by Department of Homeland Security, the International Civil Aviation Organization and other relevant organizations beforehand,"

Despite advance warnings about the 2019 GPS rollover from the Department of Homeland Security, the International Civil Aviation Organization, and other relevant organizations ..."

**Commented [SC2]:** done

>>line 42 "numerous" & "numerous"

roughly how many?

**Commented [SC3]:** At least 30, by quick count. Official numbers were not released.

>>line 58 "one for the equipment cargo and another for the field team to put in."

...another for the $N$ member field team...

**Commented [SC4]:** 3 members are usually present for a deployment, though some sites have only had two on hand for repairs.

>>line 60 maybe give a table with # of flights per site per year?

**Commented [SC5]:** The sites do not receive yearly visits, so the caches are only laid in advance of a requested visit for install or repairs.

>>line 69 "In the polar region, the available sunlight is limited throughout the year"

...limited during Southern summer as well?

**Commented [SC6]:** There are occasional drop outs likely caused by inclement weather, but on average during the southern summer we have near constant sunlight.

>>line 81 "Thus, we use this fault as an example to describe best practices for designing future systems against similar

faults, or to at least enable graceful fault recovery and restorative action if a fault occurs."

...designing against similar faults...  ==>> ...protection from similar faults...

**Commented [SC7]:** Done

>>line 85 "the remote instrument platform operated by the authors"

...the AAL-PIP...

**Commented [SC8]:** Done

>>line 89 inconsistent past/present tense?

**Commented [SC9]:** done

>>line 114 "Up to this point 0 of the 5 RUDICS enabled stations"

...none of the five...

**Commented [SC10]:** done

>>line 152 "land at PG0"

Station coordinates for PG2 provided in Figure 1, so should probably give PG0 lat/lon here.

**Commented [SC11]:** done

\>\>line 163

Is PG0 the only one able to use Iridium?  Clarify.

\>\>line 188 "all but one of the systems have come out of hibernation and each system has undergone a software update. The AAL-PIP array

has thus far been restored to full operational capacity."

N-1 of N is full capacity?

\>\>line 267 "John Bowman, and undergraduate student"

...an undergraduate student...

\>\> Figure 2 What does the dotted line indicate?

Couldn't this be reduced to a single figure just by adding 2nd line pointing down from Iridium to System clock.

=== General comments ===

Although there are some general lessons to be learned from this episode, the specific problems arose from a

particular combination of hardware and software. It might be helpful to include some technical details ie.

\>\>line 164 "GPS module"  make/model/interface?

and possibly software versions eg. nptd

The complicated sequence of events is presented over several pages, so it might be useful

to provide some summary of what happens when.  Perhaps something simple like this:
* * *
Typical state transition

0.0  system power on

0.0  SBC: start boot Linux

0.0  Iridium: boot local antenna, connect to SV

**Commented [SC12]:** All of the systems utilize the Iridium network for communications. PG1 is the only system that uses P2P Iridium, while the rest use Iridium RUDICS.

**Commented [SC13]:** At the time of writing, it was not yet expected that the remaining system would have come out of hibernation.

**Commented [SC14]:** done

**Commented [SC15]:** Added an explanation of the WD signal. A second line from iridum, a disconnecting of the SC from the WD, and a connection of the HC to the WD would be required.

0.0  GPS: start boot local antenna

1.0  Linux: NTPD startup, initialize with SBC hardware clock time

2.0  GPS: acquire SVs, initial location & time estimate

3.0  GPS: RS232 available

4.0  Linux: initialize (software?) watchdog with SBC hardware clock time(?)

5.0  Linux: connect to GPS RS232, receive NMEA time, add source to NTPD

6.0  Linux: update NTPD with GPS time **possibly wrapped**

6.1  Linux: write NTPD time to SBC hardware clock

6.2  Linux: check software watchdog?

7.0  Iridium: read time from satellite

7.1  Iridium: RS232 available

7.2  Linux: connect to Iridium network

...and so on...
* * *
And centralize just a bit more detail about each of the elements in Figure 2 with their critical interactions eg.

   SBC hardware - read by Linux software; written to on shutdown?

   Linux software - NTPD(?) version

   GPS hardware (make/model/interface=NMEA?) - default read by ntpd

   Iridium satellite network - read?

   Iridium local terminal - no read; no write?

then use these labels (SBC, NTPD, GPS etc.) to refer to different time sources consistently throughout the manuscript.

For example, "system time"  and "system clock time" and "reported time"; are these all NTPD time?

**Commented [SC16]:** The GPS module in question is the Garmin GPS 15xH, however the rollover issue has been addressed in later firmware for that module. In fact, one system was unaffected by the rollover because it wasn't built until years after the original systems, and thus benefited from the new firmware.

The systems also are unable to make use of NTPD, as GPS is only toggled on once an hour for a period of 5 minutes. Similarly, the Iridium modem is only on for 5 minutes every half hour. The method used for adjusting clock drifts is by the adjtimex command.

Though we could provide great technical detail for this fault, it remains the intent of the authors to provide a high level summary in order to focus more on general design guidelines. In that line of effort, a summary figure of the fault event sequence will be added in the final manuscript.

Again- the manuscript is certainly acceptable as is, but could possibly benefit from some relatively minor modifications.

---

## Editor Decision (ED1)

**From:** Takehiko Satoh <satoh@stp.isas.jaxa.jp>
**Cc:** satoh@stp.isas.jaxa.jp
**Date:** Fri, 21 May 2021 11:27:16 +0900

```
line 54: -82 and -85 degrees latitude
  => 82 and 85 degrees S latitude

line 153: PG0 (83.6 S, 88.6 E)
  => show two decimal places as PG2 in Figure 1 caption is so.

line 82: for designing against similar faults
  => for protection from similar faults

  (this would have been the reviewer's suggestion, I think)

line 103: 6 April 2019
  => April 6, 2019

  (this is the only place European style is used for date)

line 154: January 6th, 2020
  => January 6, 2020

line 115: none of the 5 RUDICS enabled stations
  => none of the five RUDICS enabled stations

  (the reviewer suggested "five" for "5" as well)
```

---

## Author Response (AR3)

**Associate Editor Decision: Publish subject to minor revisions (review by editor)** (21 May 2021) by Takehiko Satoh
Comments to the Author:
I appreciate the authors' great efforts to improve the manuscript which is now almost ready to be published. After going through the revised manuscript, I got the following minor points (should be very easy). Would you kindly consider incorporating these to the manuscript?

**Author's Response:**

The corresponding author having now uploaded the proper file, the authors thank the editor for their suggestions, and have made the requested revisions.